Immunoinformatic approach to design a multiepitope vaccine targeting non-mutational hotspot regions of structural and non-structural proteins of the SARS CoV2

Solanki Vandana
http://orcid.org/0000-0002-3124-5225 Tiwari Monalisa
http://orcid.org/0000-0003-1664-3871 Tiwari Vishvanath vishvanath@curaj.ac.in
Department of Biochemistry, Central University of Rajasthan , Ajmer, Rajasthan , India
Albertini Maria Cristina
Electronic publication date: 2021 Mar 23
Publication date: 2021
Volume: 9
Electronic Location ID: e11126
Received 2020 Dec 1; Accepted 2021 Feb 26
Copyright: © 2021 Solanki et al.
Copyright year: 2021
Copyright holder: Solanki et al.
License: This is an open access article distributed under the terms of the Creative Commons Attribution License, which permits unrestricted use, distribution, reproduction and adaptation in any medium and for any purpose provided that it is properly attributed. For attribution, the original author(s), title, publication source (PeerJ) and either DOI or URL of the article must be cited.
License URL: https://creativecommons.org/licenses/by/4.0/

Keywords: Multiepitope vaccine, SARS CoV2, Membrane proteins, ORF8 protein, ORF3a protein, Envelope protein, Surface glycoprotein, ORF1ab polyprotein

Funding: The authors received no funding for this work.

==============================
Background

The rapid Severe Acute Respiratory Syndrome Coronavirus 2 (SARS CoV2) outbreak caused severe pandemic infection worldwide. The high mortality and morbidity rate of SARS CoV2 is due to the unavailability of vaccination and mutation in this virus. The present article aims to design a potential vaccine construct VTC3 targeting the non-mutational region of structural and non-structural proteins of SARS CoV2.

Methods

In this study, vaccines were designed using subtractive proteomics and reverse vaccinology. To target the virus adhesion and evasion, 10 different structural and non-structural proteins have been selected. Shortlisted proteins have been screened for B cell, T cell and IFN gamma interacting epitopes. 3D structure of vaccine construct was modeled and evaluated for its physicochemical properties, immunogenicity, allergenicity, toxicity and antigenicity. The finalized construct was implemented for docking and molecular dynamics simulation (MDS) with different toll-like receptors (TLRs) and human leukocyte antigen (HLA). The binding energy and dissociation construct of the vaccine with HLA and TLR was also calculated. Mutational sensitivity profiling of the designed vaccine was performed, and mutations were reconfirmed from the experimental database. Antibody production, clonal selection, antigen processing, immune response and memory generation in host cells after injection of the vaccine was also monitored using immune simulation.

Results

Subtractive proteomics identified seven (structural and non-structural) proteins of this virus that have a role in cell adhesion and infection. The different epitopes were predicted, and only extracellular epitopes were selected that do not have similarity and cross-reactivity with the host cell. Finalized epitopes of all proteins with minimum allergenicity and toxicity were joined using linkers to designed different vaccine constructs. Docking different constructs with different TLRs and HLA demonstrated a stable and reliable binding affinity of VTC3 with the TLRs and HLAs. MDS analysis further confirms the interaction of VTC3 with HLA and TLR1/2 complex. The VTC3 has a favorable binding affinity and dissociation constant with HLA and TLR. The VTC3 does not have similarities with the human microbiome, and most of the interacting residues of VTC3 do not have mutations. The immune simulation result showed that VTC3 induces a strong immune response. The present study designs a multiepitope vaccine targeting the non-mutational region of structural and non-structural proteins of the SARS CoV2 using an immunoinformatic approach, which needs to be experimentally validated.

Introduction

Immuno-pathogenesis of the infectious epidemic COVID-19 emerged as serious intimidation worldwide. In the Coronaviridae family, among four different coronavirus classes (alpha, beta, delta and gamma), alpha and beta positive-sense RNA virus strains have been confirmed for broadly epidemic infection (De Wilde et al., 2018; Tahir ul Qamar et al., 2020). Severe Acute Respiratory Syndrome Coronavirus 2 (SARS CoV2) mediated outbreak of the disease was firstly reported in Wuhan city, China, in December 2019 (Gorbalenya et al., 2020; Huang et al., 2020; Perlman, 2020; Tahir ul Qamar et al., 2019; World Health Organization, 2020; Wu, Leung & Leung, 2020; Zhu et al., 2020). The outbreak has so far infected more than 10,00,000 patients worldwide on dated 10 May 2020. SARS CoV2 genome sequence comparison showed almost 96%, 79.5% and 40% similarities with bat coronavirus, SARS CoV and MERS CoV strain, respectively (Alamri, Tahir ul Qamar & Alqahtani, 2020; Benvenuto et al., 2020; Zhou et al., 2020). The clinical symptoms of SARS CoV2 exhibit up to 14 days in infected people with fever (≥38 °C), diarrhea, dry cough, low peripheral WBC count, respiratory disorder and low lymphocyte count (Huang et al., 2020).

Among SARS CoV2 genomic RNA, 66% codes for the non-structural or essential proteins, and 33% codes for the structural or accessory proteins. Non-structural proteins have a significant role in virus replication and transcription in the host cell, while structural proteins cover the outer surface and participate in the host evasion system to cause the infection. SARS CoV2 mutation prone (Jiayuan et al., 2020) genomic organization is a difficult task to develop a vaccine that is composed of 5′-leader-UTR-replicase-ORFab-S(Spike)-E(Envelope)-M(Membrane)-ORF6-ORF7a-ORF8-N(Nucleocapsid)-3′UTR-polyA tail. Accessory proteins like ORF3a, ORF7a, and ORF8 are essential for viral pathogenesis (Seema, 2020; Zhu et al., 2020).

To overcome the issues such as cost and time associated with traditional method (monoclonal oligonucleotides, small drug molecules), of disease prevention (Li & De Clercq, 2020), the virus was analyzed by computational methods (Chen, Liu & Guo, 2020). Recently published immuno-informatics based vaccine design against MERS virus, Ebola virus chikungunya, and Zika showed the promising potential to fight against disease (Ahmad et al., 2019; Shahid et al., 2020; Tahir ul Qamar et al., 2018, 2019). Reverse vaccinology includes different software algorithms to evaluate the immunological data that analyze the epitopes binding efficiency with human leukocyte antigen (HLA) alleles, antigenicity, allergenicity and toxicity to design the potential multiepitope subunit vaccine (De Gregorio & Rappuoli, 2012; Khan et al., 2018; Mirza et al., 2016; Patronov & Doytchinova, 2013). The final multiepitope vaccine is a group of different epitopes joined with the help of linkers that may enhance specific adaptive-immune responses in the host cell (Brennick et al., 2017; Chauhan et al., 2019; Jensen & Andreatta, 2018; Lu & Meng, 2017; Nain et al., 2020; Purcell, McCluskey & Rossjohn, 2007; Saadi, Karkhah & Nouri, 2017). To activate the host immune system, interaction of toll-like receptors (TLRs) with the designed vaccine could be a potential approach to combat viral infection. Intracellular TLRs that present on cell endosomes interact with ssRNA (TLR-7 and TLR-8), dsRNA (TLR-3) and CpG DNA (TLR-9) to activate the NFkβ mediated immune/cell response (Carty & Bowie, 2010). With protein-based epitope vaccine designing, TLR-2, TLR-4, TLR2/6 and TLR1/2 heterodimer mediated immune cell activation (IRF3/7 and NFKβ) is an essential factor to activate the innate immunity. Basal TLR-2 and TLR4 activation needs for host protective immunity, whereas its overexpression leads to subvert effect by inducing the viral cell replication in the host cell (Olejnik, Hume & Muhlberger, 2018); hence, it is better to target TLR1/2 heterodimer mediated activation signaling with mild TLR-4 interaction/activation (Olejnik, Hume & Muhlberger, 2018).

In the present study, non-mutational sequences of SARS-CoV2 were targeted to design a multiepitope vaccine construct. SARS-CoV2 proteome was investigated to identify antigenic proteins and different T-cell and B-cell epitopes were identified in selected proteins, and their antigenicity, allergenicity and physiochemical properties have been evaluated. The molecular docking of final epitope constructs was performed with different TLRs and HLA alleles to confirm the stable binding interaction of the multi-epitope vaccine-receptor complex. Vaccine constructs were further assessed for its cross-reactivity and immune response in humans, as well as their similarity with the human microbiome.

Materials and Methods

Protein sequence collection

To evaluate the coronavirus suitable antigenic vaccine target, the different protein sequences have been retrieved from NCBI (https://www.ncbi.nlm.nih.gov/). These proteins are ORF6 protein (YP_009724394.1), membrane glycoprotein (YP_009724393.1), ORF3a protein (YP_009724391.1), nucleocapsid phosphoprotein (YP_009724397.2), ORF10 protein (QHI42199.1), ORF7a protein (YP_009724395.1), envelope protein (YP_009724392.1), ORF1ab (YP_009724389.1), ORF8 protein (QHD43422.1) and surface glycoprotein (YP_009724390.1). All FASTA sequences were used as an input for further immuno-informatics analysis (Seema, 2020).

Analysis of protein antigenicity and trans-membrane helicity

In developing the chimeric multi-epitope vaccine, protein antigenicity and transmembrane helicity play a key role in vaccine successes. To evaluate the protein antigenicity, the Vaxijen server (Doytchinova & Flower, 2007) with 0.4 thresholds has been used, while trans-membrane helicity was estimated using TMHMM (Krogh et al., 2001) and Protter server (Omasits et al., 2014).

Cytotoxic T cell epitopes prediction with potential antigenicity, allergenicity and Toxicity

Shortlisted seven extracellular protein peptides were evaluated by NetCTLpan version 1.1 to predict promiscuous epitopes that can enhance the host cell’s immune response by interacting with HLA-epitope binding. NetCTLpan analysis was performed for all MHC (HLA) class I molecules supertype representative (HLA-A01:01, HLA-A02:01, HLA-A03:01, HLA-A24:02, HLA-A26:01, HLA-B07:02, HLA-B08:01, HLA-B27:05, HLA-B39:01, HLA-B40:01, HLA-B58:01, HLA-B15:01), and predicts the 8, 9, 10 and 11 mer peptides to reduce the experimental efforts. With the help of the neural network, the algorithm server has predicted promiscuous high binding affinity nonameric peptides (Stranzl et al., 2010). HLA binder peptides were further evaluated by Vaxijen (http://www.ddg-pharmfac.net/vaxijen/VaxiJen/VaxiJen.html). AllergenFP (Dimitrov et al., 2013) and Toxinpred server to confirm the epitope’s antigenicity, allergenicity and toxicity level, respectively (Gupta et al., 2013).

Immunogenicity prediction

Immunogenicity prediction (IEDB) confers the property which can elicit the cellular and humoral response in the host cell against viral infection. Shortlisted NetCTLpan epitopes were used for immunogenicity prediction that confers the property which can elicit the cellular and humoral response in the host cell against viral infection. Promiscuous epitopes ORF3a protein (one epitope), surface glycoprotein (two epitopes) and ORF1ab polyprotein (17 epitopes) were used as input peptides for immunogenicity prediction. High binding capacity epitopes (by IEDB immunogenicity tool server) were selected as positive immunogenicity epitopes. This server predicted the immunogenicity level based on the epitope positions in the expected peptide and physicochemical properties of an amino acid (Calis et al., 2013).

Helper T cell epitope prediction

With the help of the IEDB MHC II binding prediction tool, T helper cell epitopes were predicted. In vaccine development, the vaccine must cover most of the world’s population. Hence to overcome this HLA alleles variation, a reference set has been used that includes the panel of 27 alleles, which covers the >99% of the population (Greenbaum et al., 2011). IEDB MHC II server is based on a combinational library that generates the percentile rank and IC-50. The lower percentile rank represents a higher affinity of the epitope-HLA complex. The epitope-alleles affinity consensus list was generated for 15 amino acids long epitopes (Wang et al., 2008).

B cell epitope prediction

B cell epitopes detect viral infections by an antibody-based immune response. IEDB BEPIPRED (Jespersen et al., 2017) and ABCpred (Saha & Raghava, 2006) was used to analyze the B cell interacting epitopes. Overlapped epitopes from both servers were chosen for further analysis. FASTA files of all seven proteins were used as an input file. These resultant epitopes were further screened by Vaxijen, AllergenFP and TOXINPRED server to analyze the epitope’s antigenicity, allergenicity and toxicity reaction in the host cell.

Comparative cross-reactivity, IFN gamma induction analysis of MHC I, II and B cell epitopes

The comparative evaluation of all the finalized MHC I, MHC II and B cell epitopes has been done to remove the overlapping epitopes. Non-overlapping epitopes should have a unique presence in the virus. To minimize the cross-reactivity in host cells, non-overlapping epitopes were BLAST against human proteome (ID 9606) using Protein Information Resource (Chen et al., 2013). Besides, IFN gamma induction has a vital role in viral elimination and host immune response induction. Hence, we have predicted the IFN gamma induction efficiency of selected non-cross-reactive epitopes via the IFNepitope server (http://crdd.osdd.net/raghava/ifnepitope/).

Multi-epitope vaccine designing

All screened epitopes that contain the antigenic property with no allergenicity, toxicity and cross-reactivity have been finalized to design the chimeric multi-epitope vaccine construct against SARS CoV2. A potent vaccine should activate the immune response in the host cell but not activates the detrimental immunity. Hence, to maintain the balance of immunity in the host cell, half non-IFN gamma inducible peptides have been selected, and vise-versa. Epitope binding with B and T cell receptor needs a degree of freedom. Based on the linker’s length, flexibility and rigidity, EAAAK and GGGS linker have been used. EAAAK linkers rigidity is used to maintain spaces between proteins domain while GGGS linkers provide flexibility to construct with minimum linker-epitope interference to maintain the epitope function (Solanki & Tiwari, 2018).

Evaluation of potential vaccine candidate or construct

Vaccine constructs binding affinity with B and T cell, IgE antibody-associated allergic reaction and post-injection toxicity to host cell are a significant issue in vaccine development. Hence, different vaccine constructs were further investigated for antigenicity (via Vaxijen), allergenicity (via AllergenFP) and toxicity (via Toxinpred). Vaccine constructs physicochemical behavior analysis is the major key to find out the post-injection symptoms. A highly antigenic vaccine construct (i.e., VTC3) with lower allergic and toxicity has been further evaluated by the Protparam Server (Gasteiger et al., 2003). The web server provides molecular weight (KDa), theoretical isoelectric point (pI), half-life, stability index, aliphatic index, extinction coefficient and grand average of hydropathicity (GRAVY). The secondary and tertiary structure of selected vaccine construct VTC3 was predicted using PESIPRED (Buchan & Jones, 2019) and Phyre2 in intensive modeling mode (Kelley et al., 2015) respectively. To further cross-validate the model structure, ab-initio modeling of the vaccine construct was performed by RaptorX (http://raptorx.uchicago.edu). The modeled vaccine was refined by GalaxyRefine server and validated by PSVS analysis, ProSA (Wiederstein & Sippl, 2007) and QMEAN (Studer et al., 2019) per published protocol.

Molecular docking

The vaccine should induce cell-mediated and humoral immune responses in the host cell to eliminate the virus infection. VTC3 binding studies were performed with TLRs (TLR1, TLR2, TLR3, TLR4, TLR1/2, TLR6) and HLA alleles via molecular docking using PatchDock server (Schneidman-Duhovny et al., 2005). The docking was further validated by the HDOCK server (http://hdock.phys.hust.edu.cn/) (Yan et al., 2020)

Molecular dynamics simulation

The Molecular Dynamics Simulation (MDS) analysis was performed by Desmond using the OPLS3e force field and TIP3P solvent. MDS was run for 50ns in duplicate as per published methods (Tiwari, 2021). In brief, the system was built by selecting TIP3P water as a solvent; the system’s charge was neutralized by adding twenty-nine sodium ions, the system was made close to the natural system by adding 0.15M NaCl. The build system was relaxed before simulation, and MDS was performed for 50 ns. The MDS trajectories were recorded and analyzed for RMSD and RMSF.

Binding affinity and dissociation constant calculation

The binding affinity as well as dissociation constant of the VTC3 vaccine construct with the HLA and TLRs were further estimated at different simulation time periods using the PRODIGY server (Vangone & Bonvin, 2017). The MDS trajectories at different time intervals were saved in PDB format and used to calculate the binding affinity and dissociation constant.

Validation of potential vaccine candidate or construct with the human microbiome

Potential vaccine candidate sequence similarity with the gut microflora would unintentionally interfere with the proteins of the host’s microflora. To minimize this adverse effect, vaccine candidate VTC3 was BLAST against proteomes of gut flora (226 organisms) (Ramos et al., 2018) identified in the Human Microbiome Project (Peterson et al., 2009). The BLAST was performed with a significant hit (E-value ≤ 10−5) and identity ≥40% (Ramos et al., 2018). For further target prioritization, hits with identity ≥40% with a human protein were filtered out, because it could have cross-interference if the virus protein were used as a target.

Mutational sensitivity profiling of VTC3 and experimental mutation database analysis

MAESTRO web server (Laimer et al., 2016) was used for mutational sensitivity profiling of designed multiepitope vaccine construct VTC3. The amino acid sequence of VTC3 was further analyzed for any mutation present in the SARS CoV2 virus with the help of the CoV-GLUE database (update June 2020) using published methods (Tiwari, 2020b).

Immune simulations

Multi epitope vaccine immune simulation profile was performed using C-ImmSim online server (http://kraken.iac.rm.cnr.it/C-IMMSIM/). This server is based on the Celada-Seiden model that predicts the immune (humoral and cellular) stimulation against vaccine construct (Rapin et al., 2010). The simulation was performed with default parameters like random seed 12,345, simulation volume 10, simulation steps 672, and vaccine injection with no LPS. The parameters were used for vaccine injection with selected MHCs (A0101, B5802, B4403, DRB3_0101, DRB1_0101) and 1,000 antigens. The VTC3 vaccine (no LPS) injection was given on 1st day, 4 weeks and 16 weeks (equivalent time steps is 1, 84,336 steps, respectively, where each time step is 8 h). To further assess the virus clearance capacity of the active immune system, the same antigen is used as a virus and observed that virus elimination. The virus was injected at 24 weeks (504 steps) with multiplication factor 0.2, infectivity 0.6 and virus number of 1,000. The different immune responses in the host after injection of VTC3 and virus were analyzed from the different plots.

Codon adaptation for vaccine construct and in-silico cloning

The amino acid sequence of VTC3 was reverse translated to DNA. Codon optimization is important for cloning and expression of VTC3 genes. The Java Codon Adaptation tool (http://www.jcat.de/) was utilized for codon optimization of VTC3 genes in a prokaryotic organism like E. coli. The E. coli K12 strain was selected to enhance the expression efficiency of VTC3. The rho-independent transcription terminators, prokaryotic ribosome binding sites, and cleavage sites of some restriction enzymes were avoided. Sites for restriction endonuclease HindIII and BamHI were added to the N and C terminals of the VTC3 gene, respectively, and subsequently cloned between HindIII and BamHI site of pET28a (+) vector using the SnapGene 4.2 tool (https:/snapgene.com/) to ensure the in-vitro expression.

Result

Analysis of protein antigenicity and trans-membrane helicity

The workflow has been discussed in Fig. 1, which explains the reverse vaccinology mechanism used in the present study. Amino acid sequences of target proteins were collected in FASTA format to analyse antigenicity and trans-membrane helicity. All the proteins with their antigenicity score such as nucleoprotein phosphoprotein (0.50), ORF10 protein (0.71), ORF8 protein (0.65), ORF7a protein (0.64), ORF6 protein (0.61), membrane glycoprotein (0.51), an envelope protein (0.60), ORF3a protein (0.49), surface glycoprotein (0.46) and ORF1ab protein (0.46) showed their potential to exceed the immune response in the host cell. To elicit the host cell’s immune response, the transmembrane helicity of viral proteins has been evaluated. With the help of TMHMM and Protter servers, we have identified the extracellular, transmembrane and cytosolic peptides. The protein protter results showed that nucleoprotein phosphoprotein, ORF6 protein, and ORF10 protein are completely cytosolic. Hence, for further study, these three proteins were eliminated, and the rest of the seven proteins were considered for further study in designing the chimeric vaccine against SARS CoV2. The extracellular peptides of these proteins would help to interact with host PAMPs and maximize the solubility of designed vaccines.

Figure 1 Brief workflow of combinational chimeric multi-epitope vaccine designing with predicted immune cell response.

Cytotoxic T cell epitopes prediction with potential antigenicity, no allergenicity and toxicity

Cytotoxic T lymphocytes epitope predictions of all seven proteins were made using NetCTLpan 1.1 using the same HLA supertypes. Out of all seven, membrane glycoprotein (YP_009724393.1) did not get any potent HLA binder peptide. The rest six proteins showed the peptide binders with different HLA supertypes. From this server results, we have identified different epitopes ORF8 (eight epitopes), ORF7a protein (nine epitopes), envelope protein (four epitopes), ORF3a protein (three epitopes), surface glycoprotein (125 epitopes) and ORF1ab polyprotein (707 epitopes) binders. From the results, we manually screened peptides that showed a binding affinity with more than one HLA allele. This approach will minimize the HLA polymorphism so that promiscuous peptides will show binding to HLA of the wide population. After the manual screening of proteins, we have shortlisted ORF8 (one epitope), ORF7a protein (two epitopes), envelope protein (two epitopes), ORF3a protein (three epitopes), surface glycoprotein (43 epitopes) and ORF1ab polyprotein (277 epitopes) epitope binders. Based on antigenicity, allergenicity and toxicity analysis of peptide, different epitopes of ORF8 (zero epitopes), ORF7a protein (zero epitopes), envelope protein (zero epitopes), ORF3a protein (one epitope), surface glycoprotein (two epitopes) and ORF1ab polyprotein (17 epitopes) were further shortlisted, and the potential epitopes were selected that can enhance immune response via HLA activation (Table 1).

Table 1 MHC I binder epitopes showing result of antigenicity, allergenicity, toxicity and immunogenicity analysis.

S. no	Protein name	Start	Epitope	Antigenicity	Allergenicity	Toxicity	Immunogenicity	
1	ORF3a	4	MRIFTIGTV	0.69	Non-allergenic	Non-toxic	0.37	
2	Surface glycoprotein	242	WTAGAAAYY	0.63	Non-allergenic	Non-toxic	0.15	
702	FTISVTTEI	0.85	Non-allergenic	Non-toxic	−0.18	
3	Orf1ab polyprotein	3449	LSFKELLVY	0.72	Non-allergenic	Non-toxic	−0.07	
1890	EIDPKLDNY	1.61	Non-allergenic	Non-toxic	−0.2	
1502	ETISLAGSY	0.59	Non-allergenic	Non-toxic	−0.16	
3841	LSDDAVVCF	0.58	Non-allergenic	Non-toxic	0.1	
5467	YTEISFMLW	1.21	Non-allergenic	Non-toxic	−0.03	
2413	VVTTFDSEY	0.45	Non-allergenic	Non-toxic	0.1	
295	FMGRIRSVY	0.52	Non-allergenic	Non-toxic	0.125	
3471	LLDKRTTCF	1.76	Non-allergenic	Non-toxic	−0.12	
4672	SMMGFKMNY	1.3	Non-allergenic	Non-toxic	−0.26	
4615	LQAENVTGL	0.82	Non-allergenic	Non-toxic	0.19	
2166	NYMPYFFTL	1	Non-allergenic	Non-toxic	0.15	
5726	IQLSSYSLF	0.75	Non-allergenic	Non-toxic	−0.48	
524	EQKSILSPL	0.55	Non-allergenic	Non-toxic	−0.26	
5406	FELEDFIPM	1.26	Non-allergenic	Non-toxic	0.33	
724	EETGLLMPL	0.48	Non-allergenic	Non-toxic	−0.12	
4055	KLVLSVNPY	0.54	Non-allergenic	Non-toxic	−0.13	

Immunogenicity prediction

Immunogenicity prediction of ORF3a protein (one epitope), surface glycoprotein (two epitopes) and ORF1ab polyprotein (17 epitopes) epitopes had shown the positive immunogenicity score of ORF3a protein (one epitope), surface glycoprotein (one epitope) and ORF1ab polyprotein (six epitopes) epitopes (Table 1). The positive immunogenicity with HLA binding confirmed that these epitopes would elicit a high immune response.

Helper T cell epitopes prediction with antigenicity, no allergenicity and toxicity

Helper T-cell mediated 15 amino acid extended epitopes were generated against the HLA allele reference set. In results, the percentile rank 0.1 was set as the threshold to filter the HLA interacting epitopes binding affinity. Based on reference threshold, ORF8 (five epitopes), ORF7a protein (zero epitopes), membrane protein (zero epitopes), envelope protein (zero epitopes), ORF3a protein (zero epitopes), surface glycoprotein (19 epitopes) and ORF1ab polyprotein (18 epitopes) showed <0.1 percentile rank (Table 2). Low percentile epitopes antigenicity (Table 2), allergenicity and toxicity level (Table 3) further shortlisted the epitopes. The compiled results of all the analyses identified surface glycoprotein (12 epitopes), ORF1ab polyprotein (four epitopes) and ORF8 (three epitopes) as HLA-II binders with high antigenicity booster and minimum allergenicity and toxicity.

Table 2 MHC II peptide showing percentile rank, and antigenicity score.

S. no	Protein name	Allele	Start	Peptide	Percentile_rank	Antigenicity	
1	ORF8 protein	HLA-DRB3*01:01	14	QPYVVDDPCPIHFYS	0.07	0.4574	
HLA-DRB3*01:01	10	CTQHQPYVVDDPCPI	0.08	0.5165	
HLA-DRB3*01:01	13	HQPYVVDDPCPIHFY	0.08	0.55	
HLA-DRB3*01:01	12	QHQPYVVDDPCPIHF	0.08	0.8637	
HLA-DRB3*01:01	11	TQHQPYVVDDPCPIH	0.08	0.6706	
2	Surface glycoprotein	HLA-DRB1*13:02	98	KTQSLLIVNNATNVV	0.01	0.63	
HLA-DRB1*13:02	102	LLIVNNATNVVIKVC	0.01	0.09	
HLA-DRB1*13:02	100	QSLLIVNNATNVVIK	0.01	0.43	
HLA-DRB1*13:02	101	SLLIVNNATNVVIKV	0.01	0.47	
HLA-DRB1*13:02	99	TQSLLIVNNATNVVI	0.01	0.43	
HLA-DRB3*02:02	100	QSLLIVNNATNVVIK	0.02	0.43	
HLA-DPA1*01:03/DPB1*04:01	323	FGEVFNATRFASVYA	0.03	0.04	
HLA-DRB1*01:01	498	LSFELLHAPATVCGP	0.03	0.5	
HLA-DRB1*01:01	497	VLSFELLHAPATVCG	0.03	0.47	
HLA-DRB1*01:01	496	VVLSFELLHAPATVC	0.03	0.86	
HLA-DRB1*13:02	103	LIVNNATNVVIKVCE	0.03	−0.11	
HLA-DRB1*13:02	97	SKTQSLLIVNNATNV	0.03	0.62	
		HLA-DRB3*02:02	101	SLLIVNNATNVVIKV	0.03	0.47	
HLA-DRB3*02:02	99	TQSLLIVNNATNVVI	0.06	0.433	
HLA-DPA1*01:03/DPB1*04:01	324	GEVFNATRFASVYAW	0.07	−0.12	
HLA-DPA1*01:03/DPB1*04:01	322	PFGEVFNATRFASVY	0.07	0.03	
HLA-DRB1*01:01	499	SFELLHAPATVCGPK	0.09	0.2	
HLA-DRB1*01:01	495	VVVLSFELLHAPATV	0.09	0.8	
HLA-DRB3*02:02	102	LLIVNNATNVVIKVC	0.09	0.09	
HLA-DRB1*13:02	98	KTQSLLIVNNATNVV	0.01	0.63	
HLA-DRB1*13:02	102	LLIVNNATNVVIKVC	0.01	0.09	
3	Orf1ab polyprotein	HLA-DRB1*09:01	54	AIILASFSASTSAFV	0.01	0.23	
HLA-DQA1*01:02/DQB1*06:02	45	AFASEAARVVRSIFS	0.08	−0.02	
HLA-DRB1*07:01	10	CTFTRSTNSRIKASM	0.12	−0.02	
HLA-DPA1*03:01/DPB1*04:02	1	YFFTLLLQLCTFTRS	0.16	−0.37	
HLA-DRB5*01:01	33	LGRYMSALNHTKKWK	0.17	0.04	
HLA-DRB1*01:01	20	KSAFYILPSIISNEK	0.27	0.71	
HLA-DPA1*03:01/DPB1*04:02	2165	CTNYMPYFFTLLLQL	0.01	0.45	
HLA-DRB1*01:01	1801	ESPFVMMSAPPAQYE	0.01	0.54	
HLA-DRB1*09:01	474	AIILASFSASTSAFV	0.01	0.23	
HLA-DPA1*03:01/DPB1*04:02	1244	EETKFLTENLLLYID	0.03	−0.11	
HLA-DRB3*01:01	903	ATYYLFDESGEFKLA	0.04	0.23	
HLA-DQA1*01:02/DQB1*06:02	535	AFASEAARVVRSIFS	0.08	−0.02	
HLA-DRB1*15:01	747	AMPNMLRIMASLVLA	0.01	0.09	
HLA-DRB3*02:02	2720	AFVTNVNASSSEAFL	0.01	0.15	
		HLA-DRB1*11:01	865	NEFYAYLRKHFSMMI	0.02	0.22	
HLA-DQA1*05:01/DQB1*02:01	2390	QMEIDFLELAMDEFI	0.03	0.61	
HLA-DRB3*02:02	2755	NYIFWRNTNPIQLSS	0.04	0.92	
HLA-DQA1*05:01/DQB1*02:01	2393	IDFLELAMDEFIERY	0.05	0.25	
HLA-DPA1*02:01/DPB1*14:01	663	QMNLKYAISAKNRAR	0.07	1.5	

Table 3 Analysis of MHC II epitope’s allergenicity and toxicity.

S. no	Protein name	Start	Peptide	Allergenicity	Toxicity	
1	ORF8 protein	14	QPYVVDDPCPIHFYS	Allergen	–	
10	CTQHQPYVVDDPCPI	Non-allergenic	Non-toxic	
13	HQPYVVDDPCPIHFY	Non-allergenic	Non-toxic	
12	QHQPYVVDDPCPIHF	Allergen	–	
11	TQHQPYVVDDPCPIH	Non-allergenic	Non-toxic	
2	Surface glycoprotein	98	KTQSLLIVNNATNVV	Non-allergenic	Non-toxic	
100	QSLLIVNNATNVVIK	Non-allergenic	Non-toxic	
101	SLLIVNNATNVVIKV	Non-allergenic	Non-toxic	
99	TQSLLIVNNATNVVI	Non-allergenic	Non-toxic	
100	QSLLIVNNATNVVIK	Non-allergenic	Non-toxic	
498	LSFELLHAPATVCGP	Non-allergenic	Non-toxic	
497	VLSFELLHAPATVCG	Non-allergenic	Non-toxic	
496	VVLSFELLHAPATVC	Non-allergenic	Non-toxic	
97	SKTQSLLIVNNATNV	allergenic	–	
101	SLLIVNNATNVVIKV	Non-allergenic	Non-toxic	
99	TQSLLIVNNATNVVI	Non-allergenic	Non-toxic	
495	VVVLSFELLHAPATV	Non-allergenic	Non-toxic	
3	ORF1ab polyprotein	20	KSAFYILPSIISNEK	Non-allergenic	Non-toxic	
2165	CTNYMPYFFTLLLQL	allergenic	–	
1801	ESPFVMMSAPPAQYE	Non-allergenic	Non-toxic	
2390	QMEIDFLELAMDEFI	Non-allergenic	Non-toxic	
2755	NYIFWRNTNPIQLSS	Non-allergenic	Non-toxic	
663	QMNLKYAISAKNRAR	Non-allergenic	Non-toxic	

B cell epitope prediction with antigenicity, no allergenicity and no toxicity

Using the IEDB Bepipred server, different epitopes from selected proteins like ORF8 protein (two epitopes), ORF7a protein (two epitopes), membrane glycoprotein (one epitope), envelope protein (one epitope), ORF3a protein (one epitope), surface glycoprotein (28 epitopes) and ORF1ab protein (98 epitopes) have been screened. With ABCpred server-based B cell epitope analysis, the epitope of ORF8 protein (nine epitopes), ORF7a protein (seven epitopes), membrane glycoprotein (one epitope), envelope protein (three epitopes), ORF3a protein (two epitopes), surface glycoprotein (50 epitopes) and ORF1ab protein (20 epitopes) was shortlisted. To filter out the common epitope from both servers, manual screening has been done to shortlist the common overlapped epitopes. This selects different epitopes from Orf8 protein (two epitopes), ORF7a protein (two epitopes), membrane glycoprotein (one epitope), envelope protein (one epitope), ORF3a protein (one epitope), surface glycoprotein (16 epitopes) and ORF1ab protein (eight epitopes). These epitopes were selected for the further experiment (Table 4). The screened epitopes were further analyzed for antigenicity, allergenicity and toxicity (Table 5). This resulted in shortlisting of ORF8 protein (one epitope), envelope protein (one epitope), ORF3a protein (one epitope), surface glycoprotein (nine epitopes) and ORF1ab protein (six epitopes) epitopes for further study.

Table 4 Comparative analysis of B cell epitope identified using IEDB and ABCpred server and antigenicity analysis.

S. no	Protein name	Start	Epitope IEDB BepiPred	Antigenicity	ABCpred	Start	Antigenicity	
1	ORF8 protein	12	QHQPYVVDDP	0.4127	PYVVDDPCPIHFYSKW	15	0.56	
49	EAGSKSPI	0.2081	ELCVDEAGSKSPIQYI	44	−0.18	
2	ORF7a protein	18	EPCSSGTYEGNSPFHPLAD	0.39	SGTYEGNSPFHPLADN	22	0.29	
58	HVYQLRARSVSPKLFIRQE	0.59	HVYQLRARSVSPKLFI	58	0.43	
3	Membrane glycoprotein	7	TITVEELKK	0.57	DSNGTITVEELKKLLE	3	0.07	
4	Envelope protein	23	YSRVKNLNSSRVP	0.47	YVYSRVKNLNSSRVPD	21	0.54	
5	ORF3a protein	15	LKQGEIKDATPSDFVR	0.81	QGEIKDATPSDFVRAT	17	0.91	
			IGTVTLKQGEIKDATP	10	1.22	
6	Surface glycoprotein	55	VSGTNGTKRF	0.53	HRSYLTPGDSSSGWTA	230	0.6	
123	DPFLGVYYHKNNKSWMESEFRVYSSA	0.49	TVEKGIYQTSNFRVQP	292	0.67	
234	LTPGDSSSGWTA	0.68	GCLIGAEHVNNSYECD	633	0.84	
298	YQTSNFRVQP	1.18	LQSYGFQPTNGVGYQP	477	0.52	
315	PNITNLCPFGEVFNATRFASVYAWNRKRISNC	0.47	TEIYQAGSTPCNGVEG	455	−0.01	
389	GDEVRQIAPGQTGKIAD	1.06	KQIYKTPPIKDFGGFN	771	−0.22	
441	FRKSNLKPFERDISTEIYQAGSTPCNGVEGFNCYFPLQSYGFQPT	0.39	CGPKKSTNLVKNKCVN	510	0.2	
501	ELLHAPATVCGPKKSTNLVKN	0.0029	FERDISTEIYQAGSTP	449	−0.29	
619	RVYSTGSNVFQ	−0.1	SWMESEFRVYSSANNC	136	0.17	
641	VNNSYECDIPI	0.6124	EVRQIAPGQTGKIADY	391	1.38	
657	ASYQTQTNSPRRARSVASQ	0.2556	TPTWRVYSTGSNVFQT	615	0.18	
680	YTMSLGAENSVAYSNN	0.6434	VIGIVNNTVYDPLQPE	1114	0.71	
771	KQIYKTPPIKDFGGF	−0.3896	SQSIIAYTMSLGAENS	674	0.56	
792	PDPSKPSKR	0.478	VSGTNGTKRFDNPVLP	55	0.51	
1093	NFYEPQIITTD	0.36	FPNITNLCPFGEVFNA	314	0.6	
1118	VNNTVYDPLQPELDSFKEELDKYFKNHTSPDVDLGDISG	0.13	SQILPDPSKPSKRSFI	788	0.26	
			YQTQTNSPRRARSVAS	659	0.192	
7	Orf1ab polyprotein	373	CHNSEVGPEH	1.14	VVKIYCPACHNSEVGP	365	0.76	
763	LQPLEQPTSEAVEAP	0.05	TLKGGAPTKVTFGDDT	814	0.98	
813	FTLKGGAP	0.88	TSRYWEPEFYEAMYTP	3996	0.4	
1458	NLEEAAR	−0.14	AVTAYNGYLTSSSKTP	1482	0.35	
1482	AVTAYNGYLTSSSKTPEE	0.5	LNLEEAARYMRSLKVP	1457	0.33	
2241	FSSEIIGYKAI	0.26	SEAVEAPLVGTPVCIN	771	0.74	
	3072	GCSCDQLREPMLQSADAQS	0.92	CGMWKGYGCSCDQLRE	3065	0.17	
3993	NDNTSRYWEP	0.27	SSEIIGYKAIDGGVTR	2242	0.74	

Table 5 Analysis of antigenic B-Cell epitope’s allergenicity and toxicity.

S. no	Protein name	Start	Epitope IEDB bepipred	Allergenicity	Toxicity	ABCpred	Start	Allergenicity	Toxicity	
1	ORF8 protein	12	QHQPYVVDDP	Allergenic	–	PYVVDDPCPIHFYSKW	15	Non-allergenic		
2	ORF7a protein	58	HVYQLRARSVSPKLFIRQE	Allergenic	–	HVYQLRARSVSPKLFI	58	Allergen	–	
3	Membrane glycoprotein	7	TITVEELKK	Allergenic	–					
4	Envelope protein	23	YSRVKNLNSSRVP	Non-Allergenic	Non-toxic	YVYSRVKNLNSSRVPD	21	Allergen	–	
5	ORF3a protein	15	LKQGEIKDATPSDFVR	Non-Allergenic	Non-toxic	QGEIKDATPSDFVRAT	17	Allergen	–	
				IGTVTLKQGEIKDATP	10	Allergen	–	
6	Surface glycoprotein	55	VSGTNGTKRF	Non-Allergenic	Non-toxic	HRSYLTPGDSSSGWTA	230	Non-allergenic	Non-toxic	
123	DPFLGVYYHKNNKSWMESEFRVYSSA	Non-Allergenic	Non-toxic	TVEKGIYQTSNFRVQP	292	Allergen	–	
234	LTPGDSSSGWTA	Non-Allergenic	Non-toxic	GCLIGAEHVNNSYECD	633	Non-allergenic	Toxic	
298	YQTSNFRVQP	Allergenic	–	LQSYGFQPTNGVGYQP	477	Non-allergenic	Non-toxic	
315	PNITNLCPFGEVFNATRFASVYAWNRKRISNC	Allergenic	–	EVRQIAPGQTGKIADY	391	Non-allergenic	Non-toxic	
389	GDEVRQIAPGQTGKIAD	Non-Allergenic	Non-toxic	TPTWRVYSTGSNVFQT	615	Non-allergenic	Non-toxic	
641	VNNSYECDIPI	Non-Allergenic	Non-toxic	VIGIVNNTVYDPLQPE	1114	Non-allergenic	Non-toxic	
680	YTMSLGAENSVAYSNN	Non-Allergenic	Non-toxic	SQSIIAYTMSLGAENS	674	Non-allergenic	Non-toxic	
792	PDPSKPSKR	Non-Allergenic	Non-toxic	VSGTNGTKRFDNPVLP	55	Allergen	–	
				FPNITNLCPFGEVFNA	314	Allergen	–	
7	Orf1ab polyprotein	373	CHNSEVGPEH	Allergenic	–	VVKIYCPACHNSEVGP	365	Non-allergenic	Non-toxic	
813	FTLKGGAP	Non-Allergenic	Non-toxic	TLKGGAPTKVTFGDDT	814	Allergen	–	
1482	AVTAYNGYLTSSSKTPEE	Non-Allergenic	Non-toxic	TSRYWEPEFYEAMYTP	3996	Allergen	–	
3072	GCSCDQLREPMLQSADAQS	Non-Allergenic	Non-toxic	SEAVEAPLVGTPVCIN	771	Non-allergenic	Non-toxic	
				SSEIIGYKAIDGGVTR	2242	Non-allergenic	Non-toxic	

Cross-reactivity, IFN gamma induction analysis of MHC I, II and B cell epitopes

All finalized MHC I, II and B cell epitopes were used for comparative analysis (Table 6) to remove the overlapping sequences. In addition to that, cross-reactivity of selected epitopes with humans has been predicated for analyzing similarity between virus protein epitopes and host cells peptide, which eliminates autoimmune reactivity. BLAST result of all selected epitopes against the human proteome showed that no epitopes were found to have cross-reactivity reactions in the host cell. In addition to this, it was seen that in virus-human cell immune response activity, IFN gamma plays an important role. High expression of IFN gamma leads to virus clearance from human cells. IFNepitope server positive score showed the capacity of the epitope to induce the IFN secretion via T cells (TC), which was listed in Table 6. All the selected epitopes were used to design the different vaccine constructs.

Table 6 Comparative analysis of Final MHC I, MHC II and B cell epitopes.

S. no	Protein name	Start	B cell epitopes (IFNscore)	Cross-reactivity with human	START	MHC I epitopes (IFNscore)	Cross-reactivity with human	Start	MHC II Epitopes (IFNscore)	Cross-reactivity with human	
1	ORF8 protein	15	PYVVDDPCPIHFYSKW(1)	NO				10	CTQHQPYVVDDPCPI(1)	NO	
2	Envelope protein	23	YSRVKNLNSSRVP(−0.49)	NO							
3	ORF3a protein	15	LKQGEIKDATPSDFVR(0.62)	NO	4	MRIFTIGTV(−0.43)	NO				
4	Surface glycoprotein	55	VSGTNGTKRF(−0.75)	NO	242	WTAGAAAYY(0.57)	NO	98	KTQSLLIVNNATNVV(−0.37)	NO	
123	DPFLGVYYHKNNKSWMESEFRVYSSA(3.7)	NO				498	LSFELLHAPATVCGP(−0.51)	NO	
234	LTPGDSSSGWTA(−0.14)	NO	–	–	–	–	–	–	
641	VNNSYECDIPI(1)	NO	–	–	–	–	–	–	
680	YTMSLGAENSVAYSNN(−0.04)	NO	–	–	–		–	–	
792	PDPSKPSKR(0.12)	NO	–	–	–	–	–	–	
477	LQSYGFQPTNGVGYQP(14)	NO							
391	EVRQIAPGQTGKIADY(1)	NO							
1114	VIGIVNNTVYDPLQPE(12)	NO							
5	Orf1ab polyprotein	365	VVKIYCPACHNSEVGP(2)	NO	3841	LSDDAVVCF(−0.30)	NO	20	KSAFYILPSIISNEK(0.33)	NO	
771	SEAVEAPLVGTPVCIN(2)	NO	2413	VVTTFDSEY(−0.51)	NO	1801	ESPFVMMSAPPAQYE(−0.21)	NO	
2242	SSEIIGYKAIDGGVTR(2)	NO	295	FMGRIRSVY(−0.25)	NO	2390	QMEIDFLELAMDEFI(0.31)	NO	
813	FTLKGGAP(−0.22)	NO	4615	LQAENVTGL(−0.21)	NO	2755	NYIFWRNTNPIQLSS(−0.27)	NO	
1482	AVTAYNGYLTSSSKTPEE(−0.04)	NO	2166	NYMPYFFTL(−0.58)	NO	663	QMNLKYAISAKNRAR(1)	NO	
3072	GCSCDQLREPMLQSADAQS(0.41)	NO	5406	FELEDFIPM(−0.28)	NO				

Analysis of physicochemical properties of selected vaccine construct VTC3

Designed vaccine constructs VTC1, VTC2 and VTC3 (Table 7), showed antigenicity with no allergenicity and no toxicity. The physiochemical analysis of highly antigenic vaccine construct VTC3 showed that it has 227 amino acids, a molecular weight of 24 kDa, and an instability index of 23.36 that represent the protein’s stable nature. VTC3 construct contains the aliphatic and GRAVY index value of 68.72 and −0.455, respectively. PESIPRED secondary structure analysis of vaccine construct VTC3 showed the 62.56% alpha helicity, 7.05% extended strands, 5.73% beta-turn with 24.67% random coils. VTC3 tertiary structure has been modeled, refine, and validated by Ramachandran plot that showed 97% residue in the favored region (Fig. 2). The model has a Z-score of −3.35 and QMEAN of −10.75 (Fig. S1). To validate Phyre 2 model, the ab initio model was generated using RaptorX. Both models are similar, and the Ramachandran plot of the RaptoX model showed 99% residue in the favored region. This confirms that phyre2 model is valid hence used for further study.

Table 7 Sequence, Antigenicity, and Allergenicity of designed multi-epitope Vaccine constructs.

S. no	Name	Vaccine construct	Antigenicity	Allergenicity	
1	VTC1	EAAAKCTQHQPYVVDDPCPIHEYGAEALERAGYSRVKNLNSSRVPGGGSMRIFTIGTVHEYGAEALERAGKTQSLLIVNNATNVVGGGSLKQGEIKDATPSDFVRHEYGAEALERAGWTAGAAAYYGGGSLSFELLHAPATVCGPHEYGAEALERAGVSGTNGTKRFGGGSLSDDAVVCFHEYGAEALERAGKSAFYILPSIISNEKGGGSVVKIYCPACHNSEVGPEAAAK	0.51	Non-allergenic	
2	VTC2	EAAAKCTQHQPYVVDDPCPIHEYGAEALERAGYSRVKNLNSSRVPEAAAKLKQGEIKDATPSDFVRHEYGAEALERAGMRIFTIGTVEAAAKKTQSLLIVNNATNVVHEYGAEALERAGDPFLGVYYHKNNKSWMESEFRVYSSAEAAAKWTAGAAAYYHEYGAEALERAGPDPSKPSKREAAAKNYIFWRNTNPIQLSSHEYGAEALERAG SSEIIGYKAIDGGVTREAAAK LQAENVTGL	0.47	Non-allergenic	
3	VTC3	EAAAKGCSCDQLREPMLQSADAQSHEYGAEALERAGFELEDFIPMEAAAKQMNLKYAISAKNRARHEYGAEALERAGEVRQIAPGQTGKIADYEAAAKLSFELLHAPATVCGPHEYGAEALERAGWTAGAAAYYEAAAKLKQGEIKDATPSDFVRHEYGAEALERAGMRIFTIGTVEAAAKYSRVKNLNSSRVPHEYGAEALERAG PYVVDDPCPIHFYSKWEAAAK	0.61	Non-allergenic	

Figure 2 Tertiary structure of modeled VTC3 construct (A) and Ramachandran plot of the modeled proteins (B).

Molecular docking analysis confirms the interaction of VTC3 with TLR and HLA

To be effective against SARS-CoV2, a vaccine should have the capacity to activate the immune response of the human host. The virus can subvert the host protein function, which plays a role in host cell invasion or virus persistence. In a previous study, it has been shown that TLRs have an extensive role in pathogen persistence and clearance. Hence, docking analysis was performed, which showed that our construct VTC3 interacts with TLR1/2 complex, TLR1, TLR3, TLR4, TLR6 but has the highest binding affinity with TLR1/2 complex, followed by TLR4 and TLR6 (Table 8). The docked poses are shown in Fig. 3. As mentioned in the introduction, the interaction of VTC3 with TLR1/2 is important for the immune response for viral infection; hence docking results further confirm it. In addition to that, molecular docking of VTC3 with different HLAs (MHCs) has been performed, which was important to induce MHC-I and MHC-II to activate the immune response in the host cell. Best docked TLRs and HLA alleles molecules were further confirmed by the HDOCK server (Fig. S2). The binding affinity and dissociation constant of VTC3 were favorable with HLA, TLR1/2 and TLR4 (Table S1).

Table 8 Molecular docking of vaccine construct with different TLRs and HLA alleles.

S. no.	Receptor	Receptor PDB	Global Energy	
1	TLR4	2Z62	−1.64	
3UL8	2.16	
2Z63	−28.76	
3UL9	−1.43	
2Z65	−18.55	
2Z66	−23.62	
3FXI	−6.93	
3ULA	−1.75	
4G8A	−2.65	
5NAM	−6.18	
2Z64	1.23	
2	TLR1-2 hetero dimer	2Z81	−0.88	
2Z80	−31.20	
2Z82	0.22	
2Z7X	3.75	
3	TLR1	6NIH	−1.95	
1FYV	−6.49	
4	TLR2	6NIG	10.81	
5	TLR6	3A79	−13.05	
6	TLR3	2A0Z	−8.85	
1ZIW	6.11	
7	HLA Alleles	5IM7	−24.66	
1XR8	−1.95	
1SYS	−17	
1A6A	−5.59	
1A1M	−3.24	
1BX2	6.09	
1H15	−9.32	
1ZSD	5.51	
3C5J	5.12	
4O2E	9.68	

Figure 3 Docked pose of VTC3-TLR1/2 complex (A) and VTC3-TLR4 complex (B).

MDS analysis confirms the strong interaction of VTC3 with TLR1/2 heterodimer and HLA

Binding affinity from the PatchDock result showed that VTC3 has better interaction with the TLR1/2 complex than TLR4, while Binding affinity from HDOCK result showed TLR4 have better interaction than TLR1/2; hence both complexes were analyzed by MDS analysis. MDS was performed till 10 ns for both the complex, and results were analyzed for RMSD, RMSF, etc. RMSD calculation showed that the VTC3-TLR1/2 complex is stable throughout simulation with RMSD of around 3 Å (Fig. 4A), and the VTC3-TLR4 was found to be relativity less stable with RMSD of 7 Å (Fig. 4C). Similarly, the RMSF analysis showed that the VTC3-TLR1/2 complex has less fluctuation with RMSF of around 3 Å (Fig. 4B) while the VTC3-TLR4 complex showed that the vaccine has an RMSF of 7 Å (Fig. 4D). MDS result suggests that multiepitope vaccine construct VTC3 showed a stronger and stable interaction with TLR1/2 complex compared to TLR4 (Fig. 4); this indicated that this vaccine construct induces an immune response suitable for clearance of SARS CoV2. For a vaccine construct, it is also important to interact with HLA. MDS analysis of the VTC3-HLA complex was investigated till 50 ns, and the result was analyzed for RMSD and RMSF. RMSD calculation showed that the VTC3-HLA complex is stable throughout simulation with RMSD of around 5 Å at 50 ns (Fig. 5), and RMSF analysis showed that the VTC3-HLA complex has some fluctuation at the terminal. Both the data suggest the interaction between VTC3 and HLA.

Figure 4 Root-mean-square deviation and Root mean square fluctuations during molecular dynamics simulation analysis of VTC3-TLR1/2 complex (A and B) and VTC3-TLR4 complex (C and D).

Result shows a stable interaction of VTC3 with TLR1/2 complex.

Figure 5 Root-mean-square deviation during molecular dynamics simulation analysis of VTC3-HLA complex. MDS was performed till 50 ns.

Strong binding affinity and favorable dissociation constant between VTC3 with HLA and TLRs during MDS analysis

To further validate interaction, binding affinity and dissociation constant were calculated for MDS trajectories and found that VTC3 has a binding affinity of −13.1 kcal/mol at 45 ns simulation time and dissociation constant of 2.4 × 10−10 M with HLA at 45 ns. Similarity, binding affinity and dissociation constant of VTC3 was found to be favorable with TLR1/2 (−11.5 kcal/mol and 3.8 × 10−9) and TLR4 (−9.1 kcal/mol and 2.1 × 10−7) at 10 ns simulation time. This showed good interaction between VTC3 with HLA and TLR1/2. The binding affinity and dissociation constant at different simulation times (Table S2) and interacting residues of the VTC3-HLA complex (Table S3) are shown in Supplemental Data.

Multiepitope vaccine VTC3 does not have significant similarity with the human gut microbiome

NCBI blast of VTC3 construct against the proteome of 226 gut flora showed no significant similarity (Table S4) between amino acid sequence of the multiepitope vaccine construct VTC3 and amino acid sequence of the human gut microbiome that further minimize the cross-reactivity in the host cell.

Mutational profiling of multiepitope vaccine VTC3 showed less mutation sensitivity

Mutational sensitivity profiling of the VTC3 vaccine construct (Fig. 6) showed that only a few epitope residues of VTC3 have ΔΔGpred > 0, suggesting that vaccine construct VTC3 has less mutation sensitivity, which enhances the possibility of vaccine constructs to be effective. Similarly, analysis of vaccine construct VTC3 in the CoV-GLUE database showed less mutation hot spots in our vaccine construct (Table S5). This database contains all replacements, insertions and deletions, which have been observed in the GISAID hCoV-19 sequence sampled from the pandemic. Interacting residues of vaccine construct VTC3 with the HLA, TLR1/2 complex, TLR4 has been identified. The result showed that most of the residues of VTC3 with the mutation do not involve the physical interaction with these proteins. These results further support the efficacy of the vaccine construct.

Figure 6 Diagrammatic presentation of proposed combinational chimeric multi-epitope vaccine VTC3 shows different epitopes and linkers in the vaccine.

Immune simulation confirms the strong immune response against the VTC3, as well as virus protection

In the presence of antigen, the C-ImmSim model illustrates the host immune responses at both the humoral and the cellular level. The immune response against the VTC3 vaccine showed an appropriate immune response and virus antigen elimination (Fig. 7). Injection of the vaccine VTC3 without LPS shows elevated IgM level after the first injection. Its concentration further increases along with IgG1 after subsequent injection of VTC3, showing secondary and tertiary immune response. The secondary and tertiary response eliminates the antigen on a shorter timescale. It is also observed that when the virus was injected at 24 weeks (at 168 days), it is eliminated at the shortest timescale that showed the presence of memory cells to react with virus antigen. It was also observed that after virus injection, IgG1 titer is more than IgM titer (Fig. 7A). Similarly, B-cell population increases with subsequent injections of VTC3 that show a typical immune response. The memory B cell also increases after second injection and remains almost similar (Fig. 7B). The population of Th cells increases after injection of VTC3 and Th memory increase after second injection and remains almost same afterward (Fig. 7C). Among the Th cells, the population of Th1 is more dominant. After injection of VTC3, the cytotoxic TC population increases and remain constants after first injections with a TC memory (a.u. 900). It was observed that TC memory was further enhanced after injection of the virus at 168 days. It was also observed that VTC3 injection increases the IL-12 level and IFN-gamma showed a potent innate and adaptive immune response. The population of macrophages also increases after the injection of VTC3. Other cells like the dendritic cell, natural killer cells and epithelial cell population, plasma B lymphocyte population also support immune reaction. In brief, the result explains the development of strong immune memory with subsequent virus eliminating potential.

Figure 7 Immune Simulation of immunization experiment.

Injection of Vaccine VTC3 without LPS was given at 1st Day, 28th Days (4 Weeks), 112 Days (16 Weeks) and injection of the virus after 168 Days (24th week) after initial injection. (A) Antigen counts and immunoglobulins response in isotypes; (B) B-cell population observed with the administration of VTC3 and virus; (C) T helper (TH) cells population after injection of VTC3 and virus; (D) cytotoxic-T (TC) cells population after injection of VTC3 and virus. The result shows the virus antigen was eliminated at a shorter timescale due to memory cells’ presence ready to react against the epitopes.

Codon adaptation and in silico cloning

The Java Codon Adaptation tool converted the protein sequence of VTC3 into cDNA sequence. This cDNA sequence contains 52% GC content and CAI value of 1.0. This high CAI value shows a high expression of the vaccine gene in the host cell. The final vaccine construct (VTC3) was cloned into a pET28a vector between the HindIII and BamHI (Fig. 8).

Figure 8 In silico cloning of the SARS-CoV-2 vaccine in the vector, pET28a (+).

Green areas represent the COVID-19 vaccine, while the black areas represent the expression vector, pET28a (+).

Discussion

In the present study, a multiepitope vaccine constructs VTC3 for SARS CoV2 have been proposed with the help of reverse vaccinology that compiles the outer surface-exposed epitopes of the structural and non-structural proteins of this virus. There are different therapeutic strategies, and ongoing research has been tried to control SARS CoV2 (Felsenstein et al., 2020; Tiwari, 2020b, 2020a). Although different groups have tried to design a vaccine against SARS-CoV2, the present study reports a comprehensive approach to design a vaccine construct that taking care of all possible design shortcomings such as absence of surface epitopes, inappropriate TLRs interactions, less population coverage, interaction with the human microbiome, presence of cross-reactivity with human proteins, epitope form mutation hot spots and absence of immune response, etc. SARS CoV2 structural proteins like surface glycoprotein, membrane and envelope protein have a significant contribution in surface adherence and internalization, while non-structural protein is virulence-associated factors that cause immune-pathogenesis. SARS CoV2 host cell internalization is facilitated by surface glycoprotein (spike protein) interaction with the ACE2 receptor. Targeting the outer exposed peptides of both structural and non-structural protein as a vaccine target would be a promising approach to induce both humoral and cellular immune response that recognizes the virus and killed it. These proteins were shortlisted by antigenicity score and trans-membrane helicity. The shortlisted proteins were used to identify surface-exposed peptides. Surface exposed peptides of these proteins were further analyzed to determine MHC I, II and B cell-mediated epitopes. The selected epitopes were shortlisted by their antigenicity, allergenicity, toxicity, cross-reactivity, immunogenicity and IFN gamma secretion scores. Cross-reactivity of selected epitopes of the virus with the human’s protein has also predicted that eliminates autoimmune reactivity for the vaccine construct. These epitopes were used to design different vaccine constructs (VTC1, VTC2 and VTC3) with the help of linkers. Antigenicity, allergenicity and physiochemical analysis of vaccine constructs were further determined to enhance the peptide potency. For designing the reliable vaccine, a mutational hot spot site has been analyzed, and found that epitopes belong to the non-mutation site of the protein. To minimize similarity with human proteome and gut microbiome, BLASTP analysis has been done that showed no similarity of VTC3 construct with host cell proteins.

The finalized construct VTC3 should have the potential to interact with pathogen-associated molecular patterns, activates the host’s innate immune system, and consonance of the adaptive immunity via TLRs (TLR1, 2, 3, 4, 6, 7, 8 and 9). It has been observed that the TLRs mediated virus interaction not-only combat virus virulence and infection but also sometimes initiate the host system to overturn downstream signals for the benefit (replication and survival) of the virus-cell. During the viral infection, membrane-associated proteins interaction with TLR2 and TLR4 plays a nuanced role in exceeding inflammatory responses via adhesion and invasion that vitiate the host cell (Olejnik, Hume & Muhlberger, 2018). Simultaneously with TLRs, HLA alleles have a significant contribution to the activation of the robust immune response. The strong HLA-epitope interaction would maintain the signaling cascades to activate the immune response for the viral infection clearance. To prove the modeled VTC3 construct, we have docked it with different HLAs and TLRs. Presently, the focus lies on the development of vaccines against viruses that can activate the innate immune system via TLR1/2 (Carty & Bowie, 2010; Dowling & Mansell, 2016; Jensen et al., 2018) to overcome the subvert effect of the virus on TLR mediated signaling. It is reported that the CD8+ T cell-mediated HLA allele (HLA-B*5801, PDB ID 5IM7) unique interaction with immune dominant peptide contributes as a potential to control and prevent viral infection (Li et al., 2016). The interaction of VTC3 was investigated with molecular docking and MDS, and the result is in agreement that the VTC3 construct has the highest affinity with allele HLA-B*5801. It might also create a ray of hope for the potential creation of vaccines or convalescent serum antibodies against SARS CoV2. Vaccine behaviors in human is a significant concern hence immune simulation was performed that shows strong immune response after injection of VTC3, and virus elimination. The designed multiepitope vaccine construct VTC3 should be tested experimentally for therapeutic potency in future studies. Experimental validation is necessary to demonstrate the potency of the designed vaccine in future studies.

Conclusion

The high mortality and morbidity rate of COVID 19 is unprecedented due to the unavailability of vaccination; hence effective treatment strategies (inhibitor, drugs, or vaccine), rapid development, trials and production are needed immediately for this global pandemic disease. To reach out the success, it is necessary to evaluate all possible vaccine candidates to find out the one viable outcome. To increase the chance of success, WHO initiated vaccine solidarity trials to test all vaccine candidates until they fail. In the present study, an attempt was made to design a multi-epitope vaccine against SARS CoV2 that targets its exposed peptides of structural and non-structural proteins. Immunogenicity, allergenicity, toxicity and potent IFN-gamma inducer scores have also been analyzed to narrow down efficient epitopes further. Peptide matching with the human proteome showed no indication of possible cross-reactivity. However, current reverse immuno-informatics approaches were executed to target surface-exposed proteins for enhancing effective host innate with humoral and cellular immune responses. The present study concludes the design of VTC3 as a multiepitope vaccine against SARS CoV2.

Supplemental Information

Supplemental Information 1 Supplemental Figures 1–2 and Tables 1–5.

Click here for additional data file.

VT would like to thank the Central University of Rajasthan for providing the Schrodinger suite.

Additional Information and Declarations

Competing Interests

Author Contributions

Data Availability

The authors declare that they have no competing interests.

Vandana Solanki performed the experiments, analyzed the data, prepared figures and/or tables, and approved the final draft.

Monalisa Tiwari performed the experiments, authored or reviewed drafts of the paper, and approved the final draft.

Vishvanath Tiwari conceived and designed the experiments, performed the experiments, analyzed the data, prepared figures and/or tables, authored or reviewed drafts of the paper, and approved the final draft.

The following information was supplied regarding data availability:

Raw data are available as a Supplemental File.

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
