# Peer review of "Immunoinformatic approach to design a multiepitope vaccine targeting non-mutational hotspot regions of structural and non-structural proteins of the SARS CoV2"

_PeerJ, doi:10.7717/peerj.11126_

## Round 0.1 · original submission · Major Revisions

Please, consider carefully reviewer suggestions.

Reviewer 1 ·

Basic reporting

The manuscript requires thorough proof-reading to remove errors.

Experimental design

The author has carried out Immunoinformatic approach to design a chimeric multi-epitope vaccine against non-mutational hotspot region of structural and non-structural proteins of the SARS CoV2, Notwithstanding the study fails to have a novel approach and steady conclusion.

Validity of the findings

The results are not novel at any point and add no further information to the already known research in this field. The authors must make the needful incorporations before considering to communicate the article further.

• The 3D structure of construct is modeled on homology principle which even if just prediction very wrong as the construct sequence is showing very less sequence homology to the available experimental structures. The authors should use ab initio principle instead.
• The docking studies must be cross validated by another protein-peptide server to assure correctness of the binding mode with the receptor.
• Further, MMPBSA binding free energy calculations must be performed on the MD trajectories to validate the docking predictions.

Additional comments

Some serious questions need to be answered before considering this work for publication. Based on my review, I concluded the rationale of designing a vaccine when already licensed vaccines are available this work is very weak for publication.
• Why this study is performed when already licensed vaccine for SARS-COV2 is already available? This comment must be answered very clearly in the introduction section and strong rationale should be provided if any?
• How this vaccine construct is different from that many reported in the literature?

·

Basic reporting

Abstract: Need to give more reasons about the significance of your study and how it will leads to the formation of novel vaccine.

The manuscript is clear and professional english is used. The literature is well discussed.

In Paragraph 2.1, Protein sequence collection: Mention more information about the retrieval of protein sequence. For instance, the source, organisms name, incubation period, site etc.

In Paragraph, 2.12, Give more dara on which sequences of human microbiome was taken into consideration.

Experimental design

The experimental data is discussed very well.

In fig.3, Meet to add more information regarding what is the significance of red, yellow and green colour spirals shows.

Validity of the findings

The validity of findings is perfectly mentioned in the manuscript.

Additional comments

Need to check the palgriasm of the whole manuscript. Rest of all findings are well discussed.

·

Basic reporting

1- Although the manuscript is fairly easy to read and understandable, grammatical errors can be found throughout the paper (just a few examples have been highlighted in blue ). I might suggest the authors check the manuscript for all language errors and inconsistencies to reach the international audience.

2- The references are properly formatted. However, it seems references are lacking in certain sections (e.g., lines 90-95, line 180; no reference for PATCHDOCK server, line 321; should change to (Tiwari 2020a, Tiwari 2020b) as opposed to (Tiwari 2020a) (Tiwari 2020b), etc). Please check draft and add references where necessary throughout the manuscript.

3- All tables, figures, and data have been provided in good quality and scientific clarity. In particular, I would like to mention Figure 1 is very well-designed, and much-appreciated as it clarifies your approach towards chimeric protein engineering.

4- The research question is clear, and has been backed up by sufficient evidence in the main text. However, the background section of the abstract needs more explanation. It is vital to describe the rationale behind the employed in silico approach more precisely.

5- Lines 36-39: Moving such points to the methods section may improve the integrity of the abstract. The authors can start the results section by mentioning a multi-epitope vaccine was built. The results section of the abstract could as well include more precise quantitative results.

6- Lines 120-121, 137-144: The long list of HLAs is appreciated, as it increases vaccine coverage, and shows the precise reporting of study. However, it would be easier on the eye if only tabulated.

Experimental design

7- It is appreciated that, the knowledge gap has been identified properly in the title and throughout the paper as "targeting non-mutational sequences of SARS-CoV2".

8- Lines 70-73: I might add the authors mention a high affinity to ACE2 for mutational regions of the SARS-CoV2 S protein. This seemingly contradicts their rationale behind selecting non-mutational regions, in particular from the surface glycoprotein (NCBI Protein code: YP_009724390.1). I might suggest such points are include in the discussion section, as possible limitations of vaccinology against COVID-19.

9- I would like to appreciate the authors' work for Sections 2.8, 2-12, and 2.13, as these sections highlighted "cross-reactivity" and "mutation" an a vital consideration in the in silico approach towards combat against viral infections. I might encourage the authors highlight points related to these sections further in the discussion section.

10- Section 2.11: The use of up-to-date software like Desmond is appreciated, as it is a widely-accepted approach to MDS. I might suggest you explain this section with further MD settings and details. Moreover, it would be great if you could perform MM-PBSA analyses, or similar binding analyses during the MD run (see: https://www.ncbi.nlm.nih.gov/pmc/articles/PMC4487606/). If MM-PBSA analyses are not possible, you can extract MD trajectory frames every 1-2ns, and perform binding analysis of the TLR-Vaccine complex by the PRODIGY server (https://wenmr.science.uu.nl/prodigy/) or similar tools. This gives your binding analysis much more validity and significance.

11- As the methodology stands now, the methods used in the present study are valid and robust. However, the current standard of in silico vaccine research indicates the need for more explorations, as have been used by many prominent studies (just to name a few, https://www.frontiersin.org/articles/10.3389/fimmu.2020.01784/full, https://journals.plos.org/plosone/article?id=10.1371/journal.pone.0244176#, https://www.frontiersin.org/articles/10.3389/fimmu.2020.02074/full).

It is advised that the methodology is refined further in certain areas:
I. The present protein structure quality assessment method should be extended. The 3 region Ramachandran plot has been shown to be beneficial, yet inadequate for the purpose (see: https://www.ncbi.nlm.nih.gov/pmc/articles/PMC3061398/).
Please assess the designed protein quality further by ProSA tool (https://prosa.services.came.sbg.ac.at/prosa.php) or QMEAN server (https://swissmodel.expasy.org/qmean/), and provide a Z-score based validation for your protein structure as it is much more reliable than solely using a Ramachandran plot (see: https://www.ncbi.nlm.nih.gov/pmc/articles/PMC3031035/).

II. To determine the conserved regions of obtained FASTA formats, you can perform multiple alignment by online servers or BioEdit software and only select the conserved region across all proposed PubMed Protein FASTA formats.

III. You can submit your PDB structure to GalaxyRefine server (http://galaxy.seoklab.org/cgi-bin/submit.cgi?type=REFINE), and provide a comparative Ramachandran plot to indicate the protein structure has been properly refined.

IV. Please explain why your sequence does not start with "M". This may cause issues with real-world vaccine expression (see: https://pubmed.ncbi.nlm.nih.gov/20970746/). You may use SnapGene software (https://www.snapgene.com/) to conduct in silico cloning, and determine if the vaccine can be experimentally synthesized in a specific organism (e.g., E. coli).

Validity of the findings

12- The results are very valid, but may be enhanced, in particular in the area of protein quality validation. I might suggest the authors use suggested servers and add their findings to enrich the paper.

Additional comments

13- Generally, the present research is very well-conducted and can contribute to the field of in silico vaccine research. However, the methodology should be at least in part extended, in particular in the area of protein structure refinement and validation. Language and the structure of the manuscript, in particular the abstract, should be improved.

Note: All specified lines which may require enhancements have also been highlighted in yellow in the attached PDF. Also, there are many positive attributes to this work, which have been highlighted in green.

---

## Round 0.2 · accepted · Accept

The revisions have been properly performed.

·

Basic reporting

no comment

Experimental design

no comment

Validity of the findings

no comment

Additional comments

English language has been improved significantly. Also, the authors have refined their methodology to match state-of-art techniques, bringing remarkable significance to their findings. Finally, the work adds new visions into vaccine development, in particular, because the authors have taken viral mutations and cross-reactivity into account, an aspect often overlooked in the existing in silico literature. I would like to recommend acceptance and congratulate authors for their novel work.